# MODEL INVERSION ROBUSTNESS: CAN TRANSFER LEARNING HELP?

## ABSTRACT

Model Inversion (MI) attacks aim to reconstruct private training data by abusing access to machine learning models. Contemporary MI attacks have achieved impressive attack performance posing serious threats to privacy. Meanwhile, all existing MI defense methods rely on regularization that has direct conflict with the training objective, resulting in noticeable degradation in model utility. **In this work,** we take a different perspective, and propose a novel and simple method based on transfer learning (TL) to render MI-robust models. Particularly, by leveraging TL, we limit the number of layers encoding sensitive information from private training dataset, thereby degrading the performance of MI attack. We conduct an analysis using Fisher Information to justify our method. Our defense is remarkably simple to implement. Without bells and whistles, we show in extensive experiments that our method achieves state-of-the-art (SOTA) MI robustness. **Our code, pre-trained models, demo and inverted data are included in Appx.**

## 1 INTRODUCTION

Model Inversion (MI) attack is a type of privacy threat that aim to reconstruct private training data by exploiting access to machine learning models. State-of-the-art (SOTA) MI attacks (Zhang et al., 2020; Chen et al., 2021; Wang et al., 2021a; Nguyen et al., 2023) have demonstrated increased sophistication and effectiveness, achieving attack performance of over 90% in face recognition benchmarks. The implications of this vulnerability are particularly concerning in security-critical applications (Meng et al., 2021; Guo et al., 2020; Huang et al., 2020; Schroff et al., 2015; Dufumier et al., 2021; Yang et al., 2022; Dippel et al., 2021; Chang et al., 2020; Krishna et al., 2019).

The aim of our work is to propose new perspective to defend against MI attacks and to improve MI robustness. In particular, *MI robustness* pertains to the tradeoff between MI attack accuracy and model utility. MI robustness involves two critical considerations: Firstly, a MI robust model should demonstrate a significant reduction in MI attack accuracy, making it difficult for adversaries to reconstruct private training samples. Secondly, while defending against MI attacks, the natural accuracy of a MI robust model should remain competitive. A model with improved MI robustness ensures that it is resilient to MI while maintaining its utility.

**Research gap.** Despite the growing threat arising from SOTA MI, there are limited studies on defending against MI attacks and improving MI robustness. Conventionally, differential privacy (DP) is used for ensuring the privacy of individuals in datasets. However, DP has been shown to be ineffective against MI (Fredrikson et al., 2014; Zhang et al., 2020; Wang et al., 2021b). Meanwhile, a few MI defense methods have been proposed. Particularly, all existing SOTA MI defense methods are based on the idea of *dependency minimization regularization* (Wang et al., 2021b; Peng et al., 2022): they introduce additional regularization into the training objective, with the goal of minimizing the dependency between input and output/latent representation. The underlying idea of these works is to reduce correlation between input and output/latent, which MI attacks exploit during the inversion. However, reducing correlation between input and output/latent directly undermines accuracy of the model, resulting in considerable degradation in model utility (Wang et al., 2021b). To partially restore the model utility, BiDO (Peng et al., 2022) proposes to further introduce another regularization to compensate for the reduced correlation between input and latent. However, with two additional regularization along with the original training objective, BiDO requires significant effort in

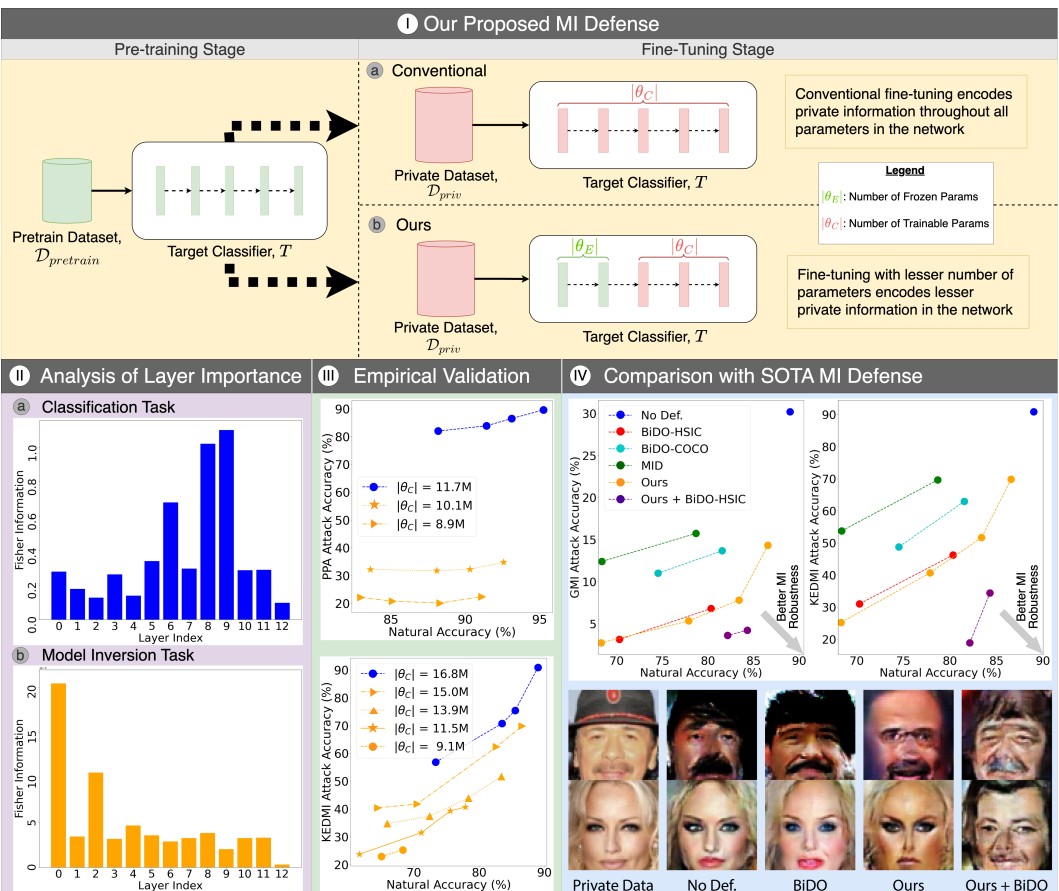

Figure 1: **(I) Our proposed MI defense (Sec. 3)**. Based on standard TL framework with pre-training (on public dataset) followed by fine-tuning (on private dataset), we propose a simple and highly-effective method to defend against MI attacks. Our idea is to limit fine-tuning with private dataset to a specific number of layers, thereby limiting the encoding of private information to these layers only (pink). Specifically, we propose to perform fine-tuning only on the last several layers. **(II) Analysis of layer importance for classification task and MI task (Sec. 4.2).** For the first time, we analyze importance of target model layers for MI. For a model trained with conventional training, we apply FI and find that the first few layers of the model are important for MI. Meanwhile, FI analysis suggests that last several layers are important for a specific classification task, consistent with TL literature (Yosinski et al., 2014). This supports our hypothesis that preventing the fine-tuning of the first few layers on private dataset could degrade MI significantly, while such impact for classification could be small. Overall, this leads to improved MI robustness. **(III) Empirical validation (Sec. 4.3).** The sub-figures clearly show that at the same natural accuracy, lower MI attack accuracy can be achieved by reducing the number of parameters fine-tuned with private dataset. **(IV) Comparison with SOTA MI Defense (Sec. 4.4).** Without bells and whistles, our method achieves SOTA in MI robustness. Visual quality of MI-reconstructed images from our model is inferior. User study confirms this finding. Extensive experiments can be found in Sec. 4.5. **Best viewed in color with zooming in.**

hyperparameter tuning based on intensive grid search (Peng et al., 2022), and is sensitive to small changes in hyperparameters (see our analysis Appx. C).

**In this paper,** our main hypothesis is that *a model with lesser parameters encoding sensitive information from private training dataset ($\mathcal{D}_{priv}$) could achieve better MI robustness.* Based on that, we propose a novel transfer learning (TL) perspective to defend against MI attacks (Fig. 1). Leveraging on standard two-stages TL framework (Pan & Yang, 2010; Yosinski et al., 2014), with pre-training on public dataset as the first stage and fine-tuning on private dataset as the second stage, we propose to limit private dataset fine-tuning only on a specific number of layers. Specifically, in the second stage, we perform private dataset fine-tuning only on the last several layers of the model. The first few layers are frozen during the second stage, preventing private information encoded in these

layers. We hypothesize that by reducing the number of parameters fine-tuned with private dataset, we could reduce the amount of private information encoded in the model, making it more difficult for adversaries to reconstruct private training data.

To justify our design, we conduct, for the first time, an analysis of model layer importance for the MI task. We propose to apply Fisher Information (FI) to quantify importance of individual layers for MI (Kirkpatrick et al., 2016; Li et al., 2020). Our analysis suggests that first few layers are important for MI. Therefore, by preventing private information encoded in the first few layers as in our proposed method, we could degrade MI significantly. Meanwhile, during pre-training, the first few layers learn low level information (edges, colour blobs). It is known that low level information is generalizable across datasets (Yosinski et al., 2014). Therefore, our proposed method has only small degrade in model utility. Overall, our proposed TL-based defense could achieve SOTA MI robustness. We remark that our method is very easy to implement. In our experiments, we apply our method to a range of models (CNN, vision transformers), see Sec. 4.5. On the contrary, BiDO has been applied to only VGG16 and ResNet-34 (Peng et al., 2022). Our contributions are:

- We propose a simple and highly effective MI defense based on TL. Our idea is a novel and major departure from existing MI defense based on dependency minimization regularization. Furthermore, while majority of TL work focuses on improving model accuracy (Pan & Yang, 2010; Jiang et al., 2022), our work focuses on degrading MI attack accuracy via TL.
- We conduct the first study to analyze layer importance for MI task via Fisher Information. Our analysis results suggest that the first few layers are important for MI, justifying our design to prevent private information encoded in the first few layers.
- We conduct empirical analysis to validate that lower MI attack accuracy can be achieved by reducing the number of parameters fine-tuned with private dataset. Our analysis carefully removes the influence of natural accuracy on MI attack accuracy.
- We conduct comprehensive experiments to show that our proposed method achieves SOTA MI robustness. As our method is remarkably easy to implement, we extend our experiments for a wide range of model architectures such as vision transformer (Tu et al., 2022), which MI robustness has not been studied before.

## 2 BACKGROUND

The target model $T$ is trained on a private training dataset $\mathcal{D}_{priv} = \{(x_i, y_i)_{i=1}^N\}$, where $x_i \in \mathbb{R}^{d_X}$ is the facial image and $y_i \in \{0, 1\}^K$ is the identity. The target classifier $T$ is a $K$-way classifier $T$: $\mathbb{R}^{d_X} \to \mathbb{R}^K$, with the parameters $\theta_T \in \mathbb{R}^{d_\theta}$. Under white-box MI, the adversary can access $T(x)$ and the $K$-dim vector of soft output. The classifier parameters $\theta_T$ are optimized using the main objective $\mathcal{L}$ as Cross Entropy loss $\mathbb{E}\left[-\log p(y_i|x_i)\right]$.

**Model Inversion Attack.** In MI attacks, an adversary exploits a target model $T$ trained on a private dataset $\mathcal{D}_{priv}$. However, $\mathcal{D}_{priv}$ should not be disclosed. The main goal of MI attacks is to extract information about the private samples in $\mathcal{D}_{priv}$. The existing literature formulates MI attacks as a process of reconstructing an input $x$ that $T$ is likely to classify into the preferred class (label) $y$. This study primarily focuses on whitebox MI attacks, which are the most dangerous, and can achieve impressive attack accuracy since the attacker has complete access to the target model. For high-dimensional data like facial images, the reconstruction problem is challenging. To mitigate this issue, SOTA MI techniques suggest reducing the exploration area to the meaningful and pertinent images manifold using a GAN. The Eq. 1 generalize step of existing SOTA whitebox MI attacks (Zhang et al., 2020; Chen et al., 2021; An et al., 2022; Struppek et al., 2022; Nguyen et al., 2023). The details for SOTA MI attacks can be found in the Appx. D.3

$$w^* = \arg\min_w (-\log P_T(y|G(w)) + \lambda \mathcal{L}_{prior}(w)) \tag{1}$$

where $-\log \mathcal{P}_T(y|G(w))$ denotes identity loss in MI attack, which guides the reconstructed $x = G(w)$ that is most likely to be classified as class $y$ by $T$. $G$ refers to generator to generate reconstructed data $x$ from latent vector $w$. The $\mathcal{L}_{prior}$ is the prior loss, which makes use of public information to learn a distributional prior through a GAN. This prior is used to guide the inversion process to reconstruct meaningful images. The hyper-parameter $\lambda$ is to balance prior loss and identity loss.

**Model Inversion Defense.** In contrast, the MI defense aims at minimizing the disclosure of training samples during the MI optimization process. First MI-specific defense strategy is MID (Wang et al., 2021b), which adds a regularization $d(x, T(x))$ to the main objective during the target classifier's training to penalize the mutual information between inputs $x$ and outputs $T(x)$. Another approach is Bilateral Dependency Optimization (BiDO) (Peng et al., 2022), which minimizes $d(x, f)$ to reduce the amount of information about inputs $x$ embedded in feature representations $f$, while maximizing $d(f, y)$ to provide $f$ with enough information about $y$ to restore the natural accuracy. However, both MID and BiDO suffer from the drawback that their regularization, i.e., $d(x, T(x))$ for MID and $d(x, f)$ for BiDO, conflict with the main training objective, resulting in an explicit trade-off between MI robustness and model utility. BiDO improves this trade-off with $d(f, y)$ but is hyperparameter-sensitive due to the optimization of three objectives, making it difficult to apply. In other words, MID and BiDO reduce MI attack accuracy by suppressing likelihood $P(y|x)$. This leads to an inevitable degrade in classification, where high likelihood $P(y|x)$ is favorable.

## 3 PROPOSED METHOD: MI DEFENSE VIA TRANSFER LEARNING

**Transfer Learning (TL).** TL (Pan & Yang, 2010; Yin et al., 2019) is an effective approach to leverage knowledge learned from a general task to enhance performance in a different task. By performing pre-training on a large general dataset and then fine-tuning on a target dataset, TL mitigates the demand for large labeled datasets, while simultaneously improving generalization and overall performance. In machine learning, TL works mostly focus on improving the model performance by adapting the knowledge to new tasks and domains (Jiang et al., 2022; Zhuang et al., 2020).

**Our proposed approach.** In contrast, our work is the first to apply TL to defend against MI attacks aiming at degrading MI attack accuracy. Therefore, our study is fundamentally different from existing TL works which aim to improve model utility (Pan & Yang, 2010; Yang et al., 2019; Kumar et al., 2022; Kamath et al., 2019; Kolesnikov et al., 2019). Our idea is to apply TL to reduce the leak of private information by limiting the number of parameters updated on private training data. Specifically, as illustrate in Fig. 1, we propose to train the target model $T$ as $T = C \circ E$ in two stages: pre-training and then fine-tuning. Particularly, in the fine-tuning stage, $E$ comprises parameters that are frozen, i.e., not updated by the private dataset $\mathcal{D}_{priv}$, while $C$ comprises parameters that are updated by $\mathcal{D}_{priv}$.

- **Stage 1: Pre-training with $\mathcal{D}_{pretrain}$.** We first pre-train $T$ using a dataset $\mathcal{D}_{pretrain}$. $\mathcal{D}_{pretrain}$ can be a general domain dataset, e.g., Imagenet, or it can be similar domain as the private dataset $\mathcal{D}_{priv}$. Importantly, $\mathcal{D}_{pretrain}$ has no class/identity intersection with $\mathcal{D}_{priv}$. Both $C$ and $E$ are updated based on $\mathcal{D}_{pretrain}$ in this stage.

- **Stage 2: Fine-tuning with $\mathcal{D}_{priv}$.** To adapt the pre-trained model from Stage 1 for $\mathcal{D}_{priv}$, we freeze $E$, i.e. parameters of $E$ are unchanged. We only update $C$ with $\mathcal{D}_{priv}$.

*We remark that pre-training has already been commonly adopted in previous works of MI attack. Therefore, in many cases, our method does not incur additional overhead (Peng et al., 2022; Nguyen et al., 2023; Chen et al., 2021; Struppek et al., 2022; An et al., 2022).* As an example, we consider the main setup of BiDO where VGG16 is used as the target classifier $T$. Following the previous works on MI attack, $T$ including $E$ and $C$ are first pre-trained on $\mathcal{D}_{pretrain}$ = Imagenet1K (Deng et al., 2009). Then, for our method, we fine-tune $C$ with $\mathcal{D}_{priv}$ = CelebA (Liu et al., 2015) while $E$ is frozen. In contrast, for other MI defense, both $E$ and $C$ are updated with $\mathcal{D}_{priv}$. We explore the design of $T$ with different number of layers updated by $\mathcal{D}_{priv}$, leading to different number of parameters in $C$ ($|\theta_C|$) updated by $\mathcal{D}_{priv}$. Using different $|\theta_C|$, we limit the amount of private information encoded in the parameters of $T$. We show that our approach improves MI robustness.

Regarding hyperparameter in our method, we determine $|\theta_C|$ by simply deciding at the *layer-level* of a deep neural network. Note that during training we use the same objective of classification task, i.e. no change in training objective is needed. Therefore, our method is much simpler and faster than SOTA MI defense BiDO (Peng et al., 2022) (see Appx. C). **In Sec. 4.2, we present our Fisher Information-based analysis to justify our method.**

Table 1: Training procedure for "no defense", existing MI defense methods (Wang et al., 2021b; Peng et al., 2022) and our method. Stage 1 (pre-training) is commonly used in existing methods to reduce the requirement for labeled datasets. Our method takes advantage of such setup to defend MI.

|         | No Defense | Existing MI defenses | Our method |
|---------|------------|----------------------|------------|
| Stage 1 | Train $T$ with standard objective on $\mathcal{D}_{pretrain}$ | | |
| Stage 2 | Fine-tune the whole $T$ with standard objective on $\mathcal{D}_{priv}$ | Fine-tune the whole $T$ with standard objective and additional *dependency minimization regularization* on $\mathcal{D}_{priv}$ | Fine-tune only $C$ with standard objective on $\mathcal{D}_{priv}$ |

## 4 EXPLORING MI ROBUSTNESS VIA TRANSFER LEARNING

We introduce the experiment setup in Sec. 4.1. In Sec. 4.2, we provide the first analysis on layer importance for MI task via Fisher Information suggesting that earlier layers are important for MI. Then, Sec. 4.3 empirically validate that MI robustness is obtained by reducing the number of parameters fine-tuned with private dataset. With the established understandings, we then compare our proposed method with current SOTA MI defenses (Wang et al., 2021b; Peng et al., 2022) in Sec. 4.4. Additionally, since our method offer higher practicality compared with the SOTA MI defenses, we expand the scope of MI defense setups to 21 MI attack setups in Sec. 4.5 and Appx. A, spanning 8 architectures, 4 private datasets $\mathcal{D}_{priv}$, 3 public datasets $\mathcal{D}_{pub}$, and 7 MI attacks.

While the above sections assume a consistent pre-trained dataset $\mathcal{D}_{pretrain}$ for the target classifier to ensure fair comparison with existing works, we also delve into novel analysis on the effect of various $\mathcal{D}_{pretrain}$ on MI robustness. We observe that *less similarity between pretrain and private dataset domains can improve defense effectiveness*. The details for this analysis can be found in Appx. A.3.

### 4.1 EXPERIMENTAL SETUP

**To ensure a fair comparison, our study strictly follows setups in SOTA MI defense method BiDO (Peng et al., 2022) in datasets, attack methods, and network architectures.** Furthermore, we also examine our defense approach with additional new datasets, recent MI attack models, and new network architectures. Note that these have not been included in BiDO. All the MI setups in our study are summarized in Tab. 2. The details for the setup can be found in Appx. D

**MI Defense Baseline.** In order to showcase the efficacy of the our proposed method, we compare our MI defense approach with several existing SOTA model inversion defense methods, which are BiDO-COCO, BiDO-HSIC (Peng et al., 2022), and MID (Wang et al., 2021b).

**Evaluation Metrics.** Following the previous MI defense/attack works, we adopt natural accuracy (Acc), Attack Accuracy (AttAcc), K-Nearest Neighbors Distance (KNN Dist), and $\ell_2$ distance metrics to evaluate MI robustness. Moreover, we also provide qualitative results and user study in the Appx. G.

### 4.2 ANALYSIS OF LAYER IMPORTANCE FOR CLASSIFICATION TASK AND MI TASK

In this section, we provide an analysis to justify our TL-based method to render MI robustness. We aim to understand importance of individual layers for MI reconstruction task, justifying our design to prevent encoding of private data information in the first few layers as an effective method to degrade MI. We study layer importance between classification and MI tasks. To quantify the importance, we compute the Fisher Information (FI) for the two tasks for individual layers.

**Fisher Information (FI) based analysis.** Fisher Information $F$ has been applied to measure the importance of model parameters for discriminative task (Kirkpatrick et al., 2016; Achille et al., 2019) and generative task (Li et al., 2020). For example, in (Kirkpatrick et al., 2016), FI has been applied to determine importance of model parameters to overcome catastrophic forgetting in continual learning. Our study extends FI-based analysis for model inversion, which has not been studied before.

Table 2: **Setups of our comprehensive experiments.** We follow the exact setups in the previous MI attacks. Following the SOTA MI defense (Peng et al., 2022), we conduct our three experiments for ResNet-34 (He et al., 2016) for VMI (Wang et al., 2021a) and VGG16 (Simonyan & Zisserman, 2014) for KEDMI/GMI (Chen et al., 2021) with $\mathcal{D}_{pub}$ = CelebA (Liu et al., 2015) and $\mathcal{D}_{priv}$ = CelebA. Furthermore, we evaluate our approach with other MI attacks (LOMMA (Nguyen et al., 2023), PPA (Struppek et al., 2022), BREPMI (Kahla et al., 2022), and MIRROR (An et al., 2022)), $\mathcal{D}_{pub}$ (FFHQ, AFHQ),$\mathcal{D}_{priv}$ (Facescrub, Stanford Dogs, VGGFace2), $T$ (IR152 (He et al., 2016), FaceNet64 (Cheng et al., 2017), Resnet-34, Resnet-18, Resnet-50 (He et al., 2016), ResNeSt-101 (Zhang et al., 2022), and MaxViT (Tu et al., 2022)). Note that these additional MI setups have not been experimented previously in the MI defense literature. In total, there are 21 MI setups spanning 7 MI attacks, 3 $\mathcal{D}_{pub}$, 4 $\mathcal{D}_{priv}$, 8 architectures of $T$, and 4 $\mathcal{D}_{pretrain}$. The experimental setups are described in more detail in the Appx. D.

| MI attack | $\mathcal{D}_{pub}$ | $\mathcal{D}_{priv}$ | $T$ | $\mathcal{D}_{pretrain}$ |
|---|---|---|---|---|
| VMI | CelebA | CelebA | ResNet-34 | None |
| KEDMI | | | VGG16 | Pubfig83/ Facescrub |
| LOMMA/BREPMI | | | | Imagenet1K |
| KEDMI/GMI | CelebA/FFHQ | CelebA | IR152/FaceNet64 | MS-CelebA-1M |
| | | | VGG16 | Imagenet1K |
| PPA | FFHQ | Facescub | ResNet-18/MaxViT | Imagenet1K |
| | AFHQ | Stanford Dogs | ResNeSt-101 | |
| MIRROR | FFHQ | VGGFace2 | ResNet-50 | |

Specifically, given a model $T$ parameterized by $\theta_T$ and input $X$, FI can be computed as (Kirkpatrick et al., 2016; Achille et al., 2019; Li et al., 2020):

$$F = \mathbb{E}\left[-\frac{\partial^2}{\partial \theta_T^2}\mathcal{L}(X|\theta_T)\right] \tag{2}$$

Here, $\mathcal{L}$ is the loss function for a particular task. Specifically, we investigate FI on classification task and MI task. For classification, we follow Achille et al. (2019) and Le et al. (2021) to use cross entropy $\mathbb{E}\left[-\log p(y_i|x_i)\right]$ as $\mathcal{L}$ and validation set $\mathcal{D}_{priv}^{val} = \{(x_i, y_i)_{i=1}^M\}$ as $X$. For MI task, we propose to use the $\ell_2$ distance between the feature representations of reconstructed images and the private image as $\mathcal{L}$:

$$\mathbb{E}\left[\left\|\Phi(\hat{x}_u^j) - \mathbb{E}\left[\Phi(x_{priv}^j)\right]\right\|_2\right] \tag{3}$$

Here, for a given input image, $\Phi$ computes the penultimate layer representation using the target model, and $\hat{x}_u^j$ is one of the MI reconstructed images for identity $j$, and $\mathbb{E}\left[\Phi(x_{priv}^j)\right]$ is the centroid feature of private image for identity $j$. Therefore, we use the distance between MI reconstructed image and private image of the same identity as the loss in FI analysis. The set of MI reconstructed images $\{\hat{x}_u^j\}_{j=1}^J$ for different identity is used as $X$. We explore different setups to compute $\mathcal{L}$, see Appx. B.1. In one setup, we perform FI analysis only at the last iteration (i.e., 3000, for the result in Fig. 1-II). As we are interested in FI at the layer level, we compute the average FI of all parameters within a layer. We use the main MI attack setup in Peng et al. (2022), i.e., VGG16 with KEDMI attack, for FI analysis.

**Observation.** The FI results in Fig. 1-II clearly suggest that the first few layers of a target model are important for MI task. Meanwhile, FI analysis suggests that the first few layers do not carry important information for a specific classification task. This observation is consistent with previous finding in work (Yosinski et al., 2014) suggesting that the earlier layers carry general features. The FI analysis justifies our design to prevent encoding of private information in the first few layers in order to degrade MI attacks, while keeping the impact on classification small. Overall, this leads to improved MI robustness. **Further results with different loss ($\ell_1$ and LPIPS (Zhang et al., 2018)) and different MI iterations can be found in Appx. B.1.**

### 4.3 EMPIRICAL VALIDATION

As shown in Fig. 1-IV, we observe a significant improvement in MI robustness when reducing the number of parameters fine-tuned with $\mathcal{D}_{priv}$. However, the relationship between MI attack accuracy and natural accuracy is strongly correlated (Zhang et al., 2020), which makes it unclear if the decrease in MI attack accuracy is due to the drop in natural accuracy.

In this section, we empirically investigate the hypothesis that *a model with fewer parameters encoding private information from $\mathcal{D}_{priv}$ has better MI robustness.* The empirical validation is reported in Fig. 1-III. Note that the number of parameters for the entire target model: $|\theta_C| = 16.8M$ for VGG16 with KEDMI setup and $|\theta_C| = 11.7M$ for Resnet-18 with PPA setup. The additional empirical validation for GMI can be found in the Appx. A.4. To separate the influence of model accuracy on MI attack accuracy, we perform PPA/KEDMI attacks on different checkpoints for each training setup, varying a wide range of natural accuracy. This is presented by multiple data points on each line.

The results clearly show that fine-tuning fewer parameters on $\mathcal{D}_{priv}$ enhances MI robustness compared with fine-tuning all parameters on $\mathcal{D}_{priv}$, regardless of the effect on natural accuracy. For instance, in the KEDMI setup, with a comparable natural accuracy of 83%, fine-tuning only $|\theta_C| = 13.9M$ reduces a third attack accuracy compared to fine-tuning $|\theta_C| = 16.8M$. The result in the PPA setup is even more supportive, where with a natural accuracy of around 91%, fine-tuning $|\theta_C| = 8.9M$ reduces the attack accuracy to 22.36% from 91.7% in $|\theta_C| = 11.7M$.

Across all configurations, we observe that the fewer parameters fine-tuned on $\mathcal{D}_{priv}$, the more robust the model. However, it is important to note that if the number of fine-tuned parameters on $\mathcal{D}_{priv}$ is insufficient, such as $|\theta_C| = 9.1M$ for KEDMI setup, the model's natural accuracy may drop drastically, rendering it unusable. Overall, our experiments strongly suggest that **better MI robustness can be achieved by reducing the number of parameters fine-tuned on $\mathcal{D}_{priv}$.**

### 4.4 COMPARISON WITH SOTA MI DEFENSE

For a fair comparison, we strictly follow the setups in SOTA MI defense (Peng et al., 2022). Specifically, we compare our approach with existing SOTA MI defenses (Wang et al., 2021b; Peng et al., 2022) against KEDMI/GMI in Fig. 1-IV. MID improves MI robustness by penalizing the mutual information between inputs and outputs during the training process, which is intractable in continuous and high-dimensional settings making MID resort to mutual information approximations rather than actual quantity (Peng et al., 2022). Therefore, we need to sacrifice significant model accuracy to observe the improvement in MI robustness, which results in a poor MI robustness compared with the SOTA defense BiDO.

Our proposed method is simple yet effective, achieving slightly better MI robustness than BiDO-HSIC without requiring additional conflict regularization, making it more feasible to recover model accuracy. We are the first to explore MI defense beyond the regularization perspective, therefore, our approach can be combined with SOTA MI defenses such as BiDO-HSIC. When combining with our approach, we strictly follow BiDO. The only difference is that BiDO is applied only to the unfrozen layers in the fine-tuning stage. The results in Fig. 1-IV show that the trade-off between utility and robustness is much improved when we combine two approach. Also, our method helps restore the utility degraded by BiDO, rendering a much more robust model (reducing MI attack accuracy by 27.36% from 46.23% to 18.87%) while improving model utility (increasing model accuracy by 1.8% from 80.35% to 82.15%). In the Appx. A.1, we provide additional comparison of our approach with BiDO and MID against VMI (Wang et al., 2021a).

### 4.5 EXTENSIVE RESULTS ON OTHER MI ATTACK SETUPS

Our proposed method is simple, easy to implement, and less sensitive to hyperparameters than BiDO, which requires intensive grid search for hyperparameter. This significant advantage allows us to extend the scope of experimental setups for the MI defense to align with the remarkable increase in MI attack setups, which are not yet evaluated in previous MI defenses (Peng et al., 2022; Wang et al., 2021b).

**Results on different $\mathcal{D}_{pub}$.** We evaluate our method against KEDMI and GMI attacks on three architectures (VGG16, IR152, FaceNet64) with varying public datasets (CelebA, FFHQ), spanning

Table 3: Our evaluation covers a wide range of MI attack setups, where the results are given in %. Specifically, we reports the MI defense results against different MI attack methods (KEDMI and GMI), as well as using different public datasets $\mathcal{D}_{pub}$ (CelebA and FFHQ), and pre-trained datasets $\mathcal{D}_{pretrain}$ (Imagenet1K and MS-CelebA-1M).

| Attack Method | $\mathcal{D}_{priv}$ | $\mathcal{D}_{pub}$ | $\mathcal{D}_{pretrain}$ | $T$ | Defense Method | $\|\theta_C\|/\|\theta_T\|$ | Acc ⇑ | Top1-AttAcc ⇓ | Top5-AttAcc ⇓ | KNN Dist ⇑ |
|---|---|---|---|---|---|---|---|---|---|---|
| KEDMI | CelebA | CelebA | ImageNet1K | VGG16 | No Def. | 16.8/16.8 | 89.00 | 90.87 ± 2.71 | 99.33 ± 0.75 | 1168 |
| | | | | | Ours | 13.9/16.8 | 83.41 | **51.67 ± 3.93** | **80.33 ± 2.91** | **1410** |
| | | | MS-CelebA-1M | IR152 | No Def. | 62.6/62.6 | 93.52 | 94.07 ± 1.82 | 99.67 ± 0.63 | 1071 |
| | | | | | Ours | 17.8/62.6 | 86.70 | **64.60 ± 4.93** | **87.67 ± 2.73** | **1333** |
| | | | | FaceNet64 | No Def. | 35.4/35.4 | 88.50 | 86.73 ± 2.85 | 98.33 ± 1.49 | 1194 |
| | | | | | Ours | 34.4/35.4 | 83.41 | **73.40 ± 4.10** | **91.67 ± 1.92** | **1265** |
| | CelebA | FFHQ | ImageNet1K | VGG16 | No Def. | 16.8/16.8 | 89.00 | 55.60 ± 3.75 | 84.67 ± 2.85 | 1407 |
| | | | | | Ours | 13.9/16.8 | 83.41 | **34.53 ± 3.43** | **65.33 ± 3.36** | **1554** |
| | | | MS-CelebA-1M | IR152 | No Def. | 62.6/62.6 | 93.52 | 70.27 ± 3.40 | 89.33 ± 2.14 | 1285 |
| | | | | | Ours | 17.8/62.6 | 86.70 | **46.53 ± 4.58** | **72.67 ± 3.16** | **1454** |
| | | | | FaceNet64 | No Def. | 35.4/35.4 | 88.50 | 57.87 ± 4.70 | 82.00 ± 3.45 | 1409 |
| | | | | | Ours | 34.4/35.4 | 83.41 | **15.27 ± 4.09** | **31.00 ± 4.24** | **1751** |
| GMI | CelebA | CelebA | ImageNet1K | VGG16 | No Def. | 16.8/16.8 | 89.00 | 30.20 ± 5.26 | 55.00 ± 5.95 | 1600 |
| | | | | | Ours | 13.9/16.8 | 83.41 | **7.80 ± 3.36** | **23.33 ± 4.60** | **1845** |
| | | | MS-CelebA-1M | IR152 | No Def. | 62.6/62.6 | 93.52 | 40.87 ± 4.76 | 66.67 ± 5.76 | 1516 |
| | | | | | Ours | 17.8/62.6 | 86.70 | **8.93 ± 3.73** | **22.67 ± 5.21** | **1819** |
| | | | | FaceNet64 | No Def. | 35.4/35.4 | 88.50 | 26.87 ± 3.75 | 49.00 ± 6.05 | 1643 |
| | | | | | Ours | 34.4/35.4 | 83.61 | **15.73 ± 4.58** | **33.00 ± 6.28** | **1752** |
| | CelebA | FFHQ | ImageNet1K | VGG16 | No Def. | 16.8/16.8 | 89.00 | 13.60 ± 4.43 | 32.00 ± 4.92 | 1725 |
| | | | | | Ours | 13.9/16.8 | 83.41 | **4.27 ± 2.56** | **12.33 ± 3.44** | **1919** |
| | | | MS-CelebA-1M | IR152 | No Def. | 62.6/62.6 | 93.52 | 24.27 ± 4.24 | 45.67 ± 6.71 | 1617 |
| | | | | | Ours | 17.8/62.6 | 86.70 | **6.13 ± 3.11** | **15.00 ± 4.98** | **1877** |
| | | | | FaceNet64 | No Def. | 35.4/35.4 | 88.50 | 13.13 ± 4.96 | 30.33 ± 5.40 | 1746 |
| | | | | | Ours | 34.4/35.4 | 83.61 | **2.60 ± 1.49** | **8.67 ± 3.64** | **2009** |

12 facial domain MI setups. These are standard setups in KEDMI/GMI, however, only 2 out of 12 setups examined in the current SOTA MI defense were presented in (Peng et al., 2022). The result in Tab. 3 demonstrate that our approach consistently achieves significantly more robust models across all setups while maintaining acceptable natural accuracy, with significant improvements in robustness across a wide range of attack scenarios (13.33%-42.60% for KEDMI, 11.14%-31.94% for GMI). *On average, our method with the decrease of 5.77% in natural accuracy, however, it significantly reduces the accuracy of MI attacks by more than half.*

**Results on SOTA MI attacks**. Given the remarkable advancements in MI attack research, we also provide our defense results against SOTA MI attacks (Nguyen et al., 2023; Kahla et al., 2022) on both $64 \times 64$ images (Tab. 7) and $224 \times 224$ images (Struppek et al., 2022; An et al., 2022) (Tab. 4). To the best of our knowledge, our work is the first MI defense approach against such high resolution MI attack. When addressing low-resolution MI attacks in Tab. 7, all existing defenses have suffered in natural accuracy, and our method has suffered the least in natural accuracy while reducing the most in attack accuracy. Consequently, our method achieves the best MI robustness trade-off, which can be quantified by the ratio of drop in attack accuracy to drop in natural accuracy (the larger is the ratio, the better is MI robustness trade-off). In the context of high-resolution MI attacks in Tab. 4, the results are even more encouraging. We observe only a small reduction in natural accuracy, while the attack accuracy experiences a significant drop. Additional MI defense results against BREPMI (Kahla et al., 2022) can be found in Appx. A.2.

**Results on different architectures of $T$**. As discussed, our approach is architecture-agnostic and does not required an intensive grid search for hyperparameters selection for one particular architecture. Therefore, our approach offer a higher practicability to other architectures compared with SOTA MI defense (Peng et al., 2022). In addition to the standard VGG16 architecture, we conducted evaluations on a range of other architectures, including residual-based networks such as ResNet-18, ResNet-50, ResNeSt-101, IR152, as well as the more recent MaxViT architecture (Tu et al., 2022). Across all these experiments in Tab. 4 and Tab. 7, our proposed MI defense consistently demonstrated superior performance, highlighting its effectiveness and versatility across various architectures.

Table 4: Empirical results for current SOTA MI attacks on 224x224 images. We strictly follow experimental setups from PPA and MIRROR, , where the results are given in %. Our approach successfully defends against SOTA MI attacks on high resolution 224x224. To train our defense models, we set $|\theta_C|$ = 8.9M/ 18.3M/ 27.9M/ 32.9M for $T$ = ResNet-18/ MaxViT/ ResNeSt-101/ ResNet-50, respectively.

| Attack Method | $\mathcal{D}_{priv}$ | $T$ | Defense | Acc ⇑ | AttAcc ⇓ | $\delta_{Eval}$ ⇑ | $\delta_{FaceNet}$ ⇑ | $\ell_2$ Dist ⇑ | FID ⇑ |
|---|---|---|---|---|---|---|---|---|---|
| PPA | Facescrub | ResNet-18 | No Def. | 94.22 | 88.46 | 123.85 | 0.7441 | - | 41.73 |
| | | | **Ours** | **91.12** | **22.36** | **167.44** | **1.0229** | **-** | **53.71** |
| | | MaxViT | No Def. | 96.57 | 79.63 | 128.46 | 0.7775 | - | 50.37 |
| | | | **Ours** | **93.01** | **21.17** | **168.85** | **1.0199** | **-** | **55.50** |
| | Stanford Dogs | ResNeSt-101 | No Def. | 75.07 | 91.90 | 62.56 | - | - | 33.69 |
| | | | **Ours** | **79.54** | **60.88** | **83.57** | **-** | **-** | **46.01** |
| MIRROR | VGGFace2 | ResNet-50 | No Def. | 99.44 | 84.00 | - | - | 602.41 | - |
| | | | **Ours** | **99.40** | **50.00** | **-** | **-** | **650.28** | **-** |

Table 5: Empirical results for current SOTA MI attacks on 64x64 images, where the results are given in %. Following the exact experimental setups from LOMMA, $\mathcal{D}_{priv}$ = CelebA, $\mathcal{D}_{pub}$ = CelebA, evaluation model = FaceNet, and target classifier $T$ = VGG16, there are a total of 300 attack classes. Our approach achieves better trade-off between Acc and AttAcc. $\Delta AttAcc$ and $\Delta Acc$ are computed by comparing to No Def.

| Defense | Acc ⇑ | AttAcc ⇓ | $\frac{\Delta AttAcc}{\Delta Acc}$ ⇑ | KNN ⇑ |
|---|---|---|---|---|
| No Def. | 89.00 | 95.67 | - | 1158.27 |
| BiDO | 80.35 (-8.65) | 70.47 (-25.20) | 2.91 | 1293.25 |
| Ours $|\theta_C|$ = 13.9M | 83.41 (-5.59) | 75.67 (-19.67) | **3.58** | 1303.65 |
| Ours $|\theta_C|$ = 11.5M | 78.86 (-10.14) | 59.68 (-35.99) | **3.54** | 1370.67 |

**Result on different $\mathcal{D}_{priv}$.** While the SOTA MI defense (Peng et al., 2022) primarily concentrates on the facial dataset CelebA as $\mathcal{D}_{priv}$, we extend our examination on large-scale facial datasets, such as Facescrub (Ng & Winkler, 2014) and VGGFace2 (Cao et al., 2018) Furthermore, we go beyond the facial domain by studying on the animal domain, i.e., Stanford Dogs dataset (Khosla et al., 2011). Via our comprehensive evaluation, we find that our approach consistently demonstrates its efficacy across various datasets, regardless multiple factors such as the number of training/attack classes or the specific domain under consideration. This versatility highlights the robustness and adaptability of our MI defense method across a wide range of scenarios.

In conclusion, all these extensive results consistently support that our method is effective in defending against advanced MI attacks with minimal changes to the original training of target classifier $T$.

## 5 CONCLUSION

In this paper, we propose a simple and highly effective MI defense based on transfer learning (TL). Our method is a major departure from existing MI defense based on dependency minimization regularization. Our main idea is to leverage TL to limit the number of layers encoding private data information, thereby degrading the performance of MI attacks. To justify our method, we conduct the first study to analyze layer importance for MI task via Fisher Information. Our analysis results suggest that the first few layers are important for MI, justifying our design to prevent private information encoded in the first few layers. Our method is remarkably simple to implement. Through extensive experiments, we demonstrate SOTA effectiveness of our approach across 21 MI setups spanning 8 architectures, 4 private datasets $D_{priv}$, and 7 MI attacks.

**Limitation.** Following other MI attack/defense research, our focus is on classification. Our future work studies MI attack and defense for other machine learning tasks, e.g. object detection.

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

APPENDIX

In this appendix, we provide additional experimental results in Appx. A, additional analysis in Appx. B, limitation of existing MI defenses in Appx. C, detailed experiment setting in Appx. D, reproducibility details in Appx. E and Appx. F, and qualitative result in Appx. G. These are not included in the main paper due to space limitations. The PyTorch code, a demonstration, and pre-trained models can be accessed at the following anonymous links:

- Code and pre-trained: available here
- Demonstration: available here
- Inverted data: available here

## A ADDITIONAL RESULTS

### A.1 ADDITIONAL COMPARISON WITH SOTA MI DEFENSE ON VMI

In addition to the comparison with KEDMI and GMI in Sec. 4.4, following BiDO setups, we also report the defense results agaisnt VMI in Tab. 6.

MID require to modify the last layer of the $T$ to implement the variational approximation of the mutual information (Peng et al., 2022). Hence, we observe a significant drop in natural accuracy when applying MID to VMI. BiDO partially addresses this problem, where BiDO recovers natural accuracy better with comparable attack accuracy with MID. On the other hand, our approach updates $|\theta_C| = 21.14M$ parameters out of the 21.5M parameters and improves natural accuracy by around 1% to 3% while achieving greater robustness by reducing attack accuracy by around 6%.

Table 6: MI performance against VMI. We use $T$ = ResNet-34, $\mathcal{D}_{priv}$ = CelebA.

|            | Acc ⇑  | Top1-AttAcc ⇓      | Top5-AttAcc ⇓      |
|------------|--------|--------------------|--------------------|
| No Def.    | 69.27  | $39.40 \pm 22.70$  | $61.50 \pm 21.24$  |
| MID        | 52.52  | $29.05 \pm 23.99$  | $51.05 \pm 28.52$  |
| BiDO-COCO  | 59.34  | $29.45 \pm 16.54$  | $54.25 \pm 21.01$  |
| BiDO-HSIC  | 61.14  | $30.25 \pm 23.46$  | $53.35 \pm 21.01$  |
| **Ours**   | **62.20** | **$23.70 \pm 21.38$** | **$45.95 \pm 26.10$** |

### A.2 ADDITIONAL RESULT ON BREPMI

Table 7: Empirical results for BREPMI (Kahla et al., 2022) attacks on 64x64 images. Following the exact experimental setups from BREPMI, $\mathcal{D}_{priv}$ = CelebA, $\mathcal{D}_{pub}$ = CelebA, evaluation model = FaceNet, and target classifier $T$ = VGG16, there are a total of 300 attacked classes. Our approach achieves better trade-off between Acc and AttAcc. $\Delta AttAcc$ and $\Delta Acc$ are computed by comparing to No Def.

| Defense  | Acc ⇑          | AttAcc ⇓         | $\frac{\Delta AttAcc}{\Delta Acc}$ ⇑ | KNN ⇑    |
|----------|----------------|------------------|-------------------------------------|----------|
| No. Def  | 89.00          | 69.67            | -                                   | 1337.01  |
| BiDo     | 80.35 (-8.65)  | 39.73 (-29.94)   | 3.46                                | 1534.48  |
| Ours     | 83.41 (-5.59)  | 42.00 (-27.67)   | **4.95**                            | 1517.38  |

### A.3 Effect of $\mathcal{D}_{pretrain}$ to MI Robustness

In these above sections, we use a consistent and standard pre-trained dataset to ensure fair comparison with other methods in the literature. Since the pre-trained backbone can be produced with different datasets in practice, we investigate the impact of different pre-trained datasets on MI robustness in this section. Specifically, we implement the same setup as the KEDMI setup outlined in Sec. 4.3, but vary three different pre-trained datasets: ImageNet1K, Facescrub, and Pubfig83. The results are shown in Fig. 2.

Updating all parameters $|\theta_C|$ = 16.8M on $\mathcal{D}_{priv}$, yields no significant differences among different $D_{pretrain}$. This is expected and align with our understanding, where all the parameters in $T$ are exposed to private data during the training of $T$. With fewer trainable parameters on $\mathcal{D}_{priv}$, we notice clearer differences. Overall, pre-training on a closer domain (Pubfig83 and Facescrub) restores natural accuracy much better than pre-training on a general domain (Imagenet1K).

For instance, with $|\theta_C|$ = 2.1M, pre-training on Facescrub and Pubfig83 achieve 81.48% and 69.41% accuracy, respectively, compared to 29.59% in the Imagenet1K setup. Nevertheless, pre-training on a closer domain also increases the risk of MI attack. As those frozen parameters during the fine-tuning on $D_{priv}$ keep the feature representations from $\mathcal{D}_{pretrain}$, thus, the closer the $\mathcal{D}_{pretrain}$, the riskier it is for the model against MI attack. Notably, with $|\theta_C|$ = 15.0M, models pre-training on ImagNet1K and Pubfig83 achieve comparable accuracy. However, using ImageNet1K as $\mathcal{D}_{pretrain}$ renders a more robust model (decreasing MI attack accuracy by 8.06%) than the setup of Pubfig83. In conclusion, when using our approach to train a MI robust model, it is critical to choose the $\mathcal{D}_{pretrain}$ for a trade-off between restoring model utility and robustness. Specifically, **less similarity between pretrain and private dataset domains can improve defense effectiveness.**

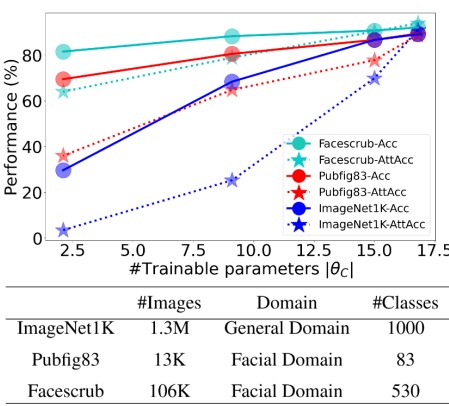

| | #Images | Domain | #Classes |
|---|---|---|---|
| ImageNet1K | 1.3M | General Domain | 1000 |
| Pubfig83 | 13K | Facial Domain | 83 |
| Facescrub | 106K | Facial Domain | 530 |

Figure 2: The effect of different $D_{pretrain}$, i.e., ImageNet1K, Pubfig83, and Facescrub. We use $T$ = VGG16, $\mathcal{D}_{priv}$ = CelebA.

### A.4 Additional Empirical Validation on GMI

Beside the empirical validation on VGG16 with KEDMI and Resnet-18 with PPA presented in the Sec. 4.3, we also provide additional empirical validation on VGG16 with GMI in Fig. 3. The observation is consistent with the result in Fig. 1-III.

## B Additional Analysis

### B.1 Additional Analysis of Layer Importance

**FI across MI iterations.** MI is a multiple iteration process. The FI for MI in Figure 1 is computed at the last iteration (the iteration that we present the result throughout our submission). Figure 5 also provides the FI across multiple iterations. We observe that after a few iterations, the FI for earlier layers keeps dominant compared to the later layers.

**Different MI losses.** In Section 4.2, we use $l_2$ distance to compute the MI loss. In addition, we provide FI results using $l_1$ distance and LPIPS (Zhang et al., 2018) to compute the MI loss. The FI results obtained using different MI loss functions are consistent with our main FI observation in Figure 1-II.

These additional FI results are consistent with those in our main FI observation in Fig. 1-II.

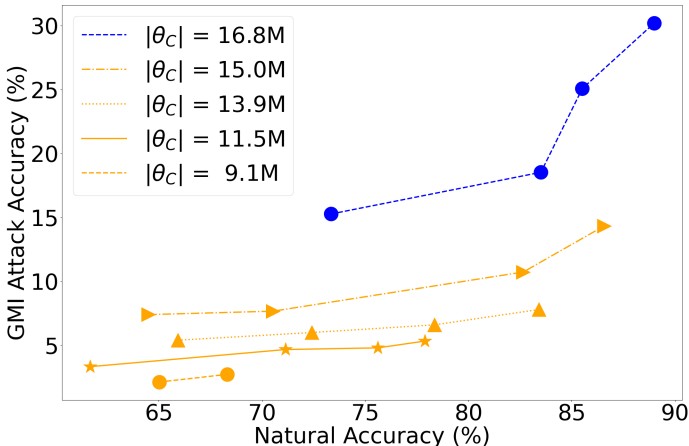

Figure 3: Empirical Validation on VGG16 with GMI. Each line represents one training setup for $T$ with a different $|\theta_C|$ updated on $\mathcal{D}_{priv}$. Note that number of parameters for the entire target model $|\theta_T| = 16.8M$ for this MI setup. To separate the influence of natural accuracy on MI attack accuracy, we perform GMI attacks on different checkpoints for each training setup, varying a wide range of natural accuracy. This is presented by multiple data points on each line. For a given natural accuracy, it can be clearly observed that attack accuracy can be reduced by decreasing $|\theta_C|$, i.e., decreasing parameters updated on $\mathcal{D}_{priv}$.

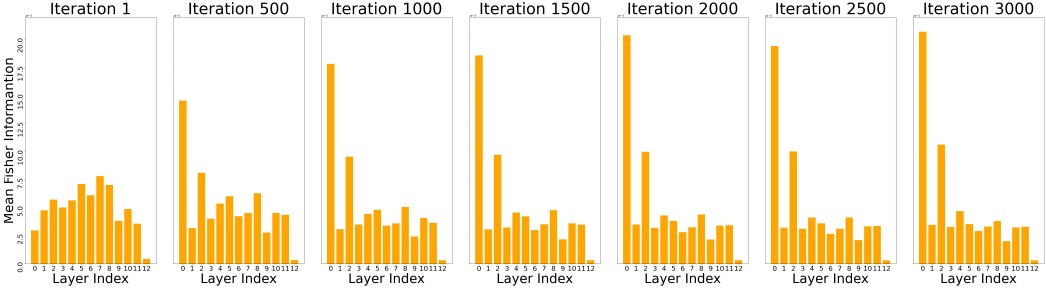

Figure 4: FI distributions across layers during all MI steps

## B.2 MI ROBUSTNESS VIA THE FALSE POSITIVE CONCEPT

We provide additional analysis in this Appendix to provide a clear understanding of how our proposed method effectively defends against MI attacks, leading to more false positive during MI attacks and decrease in attack accuracy.

As discussed, it has been shown that when a deep neural network-based classifier, denoted as $T = C \circ E$, is pre-trained on a large-scale dataset $\mathcal{D}_{pretrain}$, the features learned in the earlier layers $E$ are transferable to another somewhat related classifier on datasets $\mathcal{D}_{priv}$, enabling the model to maintain its natural accuracy without explicitly updating its parameters on $D_{priv}$ in the earlier layers (Yosinski et al., 2014). This transferability of features benefits our proposed method through maintaining the model classification performance and natural accuracy.

In contrast, MI attacks require accurate features to reconstruct the private training dataset $\mathcal{D}_{priv}$. By refraining from updating $E$ on $\mathcal{D}_{priv}$, we limit the leakage of private features into $E$, thereby improving MI robustness. Specifically, recall MI attacks are usually formulated as:

$$w^* = \arg\min_w (-\log P_T(y|G(w)) + \lambda \mathcal{L}_{prior}(w)) \tag{4}$$

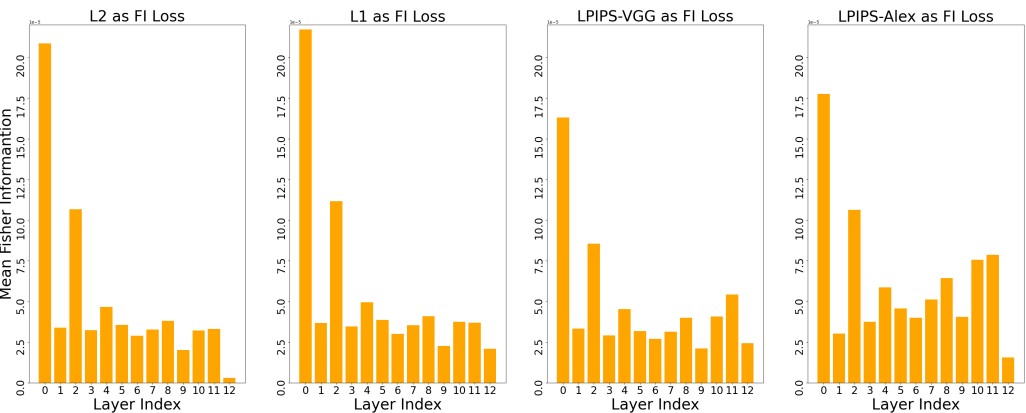

Figure 5: FI distributions across layers via different FI losses

Therefore, MI attacks aim to seek $w$ with high likelihood $P_T(y|G(w))$. We make this key observation to understand how our proposed transfer learning (TL) can degrade MI task: *With our proposed transfer learning as defense, while latent variables with high likelihood $P_T(y|G(w))$ can still be identified via Eq. 4, many $w^*$ are false positives, i.e. $G(w^*)$ do not resemble private samples. This results in decrease in attack accuracy.* This can be observed from the likelihood distributions $P_{T_{|\theta_C|=16.8M}}$ and $P_{T_{|\theta_C|=13.9M}}$ for both KEDMI (see Fig. 6) and GMI (see Fig. 7), which are similar and close to 1. These findings indicate that with our defense model, Eq. 4 could still perform well to seek latent variables $w$ to maximize the likelihood $P_T(y|G(w))$. However, although likelihood distributions $P_{T_{|\theta_C|=16.8M}}$ and $P_{T_{|\theta_C|=13.9M}}$ are similar under attacks, the attack accuracy of model with $|\theta_C| = 13.9M$ is significantly lower than that with $|\theta_C| = 16.8M$. This suggests that, due to lack of private data information in $E$ in our TL based model $|\theta_C| = 13.9M$, many $w^*$ do not correspond to images resembling private images.

In the setup where $|\theta_C| = 16.8M$, the optimization process causes the latent variables $w$ to converge towards regions that are closer to the private samples. This outcome is expected since the model possesses richer low-level features from the private dataset $D_{priv}$ in both $E$ and $C$. Consequently, we observe more true positives after MI optimization, where the likelihood $P_T(y|G(w))$ is well maximized, and the evaluation model successfully classifies them as label $y$.

In contrast, in the setup where $|\theta_C| = 13.9M$, the lack of low-level features from $\mathcal{D}_{priv}$ in $E$ hinders the optimization process. As a result, we observe a higher number of false positives after MI optimization. Although these instances successfully maximize the likelihood $P_T(y|G(w))$, the evaluation model is unable to classify them as label $y$ correctly. Therefore, this behavior indicates a higher level of robustness against the MI attack.

## C    THE LIMITATION OF EXISTING MI DEFENSE WORKS

**Conflicting objectives between classification and MI defense regularizers:** One limitation of the existing MI defenses (Wang et al., 2021b; Peng et al., 2022) is the introduction of additional regularizers that conflict with the primary objective of minimizing the classification loss (Peng et al., 2022). This conflict often leads to a significant decrease in the overall model utility.

**BiDO is sensitive to hyper-parameters**. BiDO (Peng et al., 2022), while attempting to partially recover model utility, suffers from sensitivity to hyper-parameters. Optimizing three objectives simultaneously is a complex task, requiring careful selection of weights to balance the three objective terms. The Tab. 8 results in an explicit accuracy drop when adjusting hyper-parameters $\lambda_x$ and $\lambda_y$ even with a small change. The optimized values for $\lambda_x$ and $\lambda_y$ in BiDO are obtained through a grid search (Peng et al., 2022). For example, in the case of BiDO-HSIC, the authors tested values of $\lambda_x \in [0.01, 0.2]$ and $\frac{\lambda_y}{\lambda_x} \in [5, 50]$. Furthermore, BiDO requires an additional parameter, $\sigma$, for

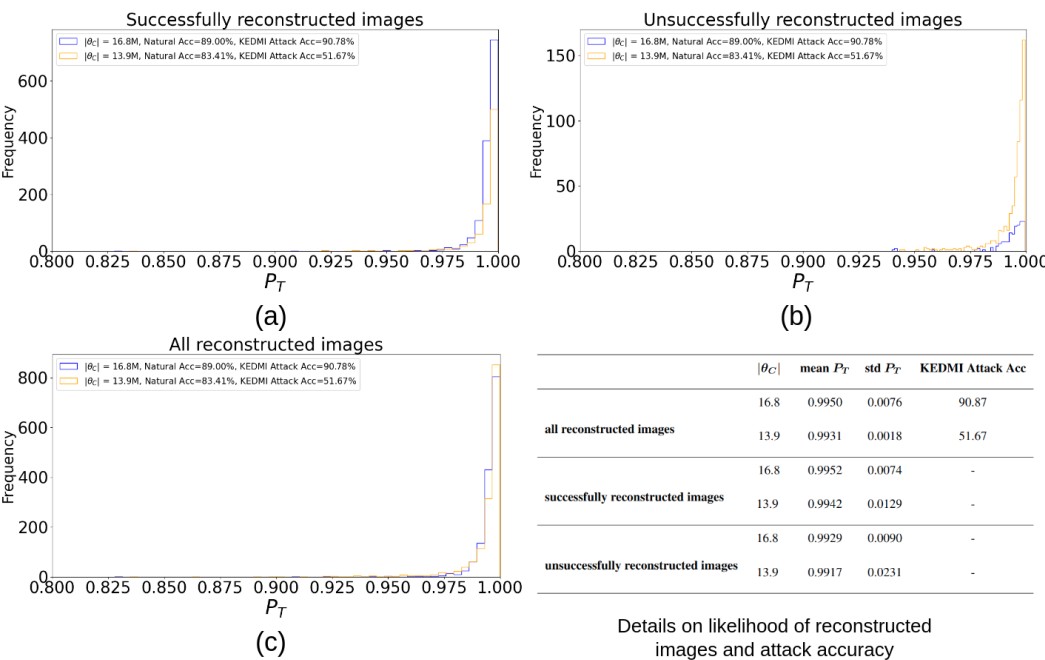

Figure 6: Visualization of the distribution of $P_T$ for two models: the no defense model (with $|\theta_C| = 16.8M$) and our proposed approach (with $|\theta_C| = 13.9M$). The visualization is conducted using KEDMI as the attack method, with $\mathcal{D}_{priv}$ = CelebA, $\mathcal{D}_{pub}$ = CelebA, $\mathcal{D}_{pretrain}$ = Imagenet1K, and $T$ = VGG16. We observe that both our defense model and the model without defense exhibit similar distributions of $P_T$. The values of $P_T$ for both successfully and unsuccessfully reconstructed images are very close to 1 in both cases. However, the attack accuracy shows a significant drop from 90.87 to 51.67 when our defense mechanism is applied.

Table 8: The SOTA MI defense, BiDO is sensitive to hyper-parameters, posing challenges for applying effectively to different architectures of target classifier $T$ or private dataset $D_{priv}$. BiDO simultaneously optimizes two objectives: $d(x, f)$ (limiting information of input $x$ and feature representations $f$) and $d(f, y)$ (providing sufficient information about label $y$ to $f$), in addition to the main objectives $\mathcal{L}$. Therefore, the final objective is $\mathcal{L} + \lambda_x d(x, z) + \lambda_y d(f, y)$, where careful weight selection for $\lambda_x$ and $\lambda_y$ is necessary to achieve a balanced training among three objectives. It is clear that inappropriate values of $\lambda_x$ and $\lambda_y$ in BiDO cause an unstable training $T$. *Note that (Peng et al., 2022) requires an extensive grid to determine suitable values for $\lambda_x$ and $\lambda_y$*

| $\lambda_x$ | 0.05 | 0.05 | 0.05 | 0.06 | 1.0 | 1.0 | 1.0 |
|---|---|---|---|---|---|---|---|
| $\lambda_y$ | 0.5 | 0.4 | 0.6 | 0.5 | 0.5 | 5.0 | 10.0 |
| Acc | 80.35 | 73.69 | 76.46 | 76.13 | 23.27 | 57.57 | 57.04 |

applying Gaussian kernels to inputs $x$ and latent representations $z$ in order to utilize COCO (Gretton et al., 2005b) and HSIC (Gretton et al., 2005a) as dependency measurements.

# D EXPERIMENT SETTING

## D.1 DETAILED EXPERIMENTAL SETUP

**Attack Dataset.** Following existing MI works (Zhang et al., 2020; Wang et al., 2021a; Chen et al., 2021; Nguyen et al., 2023), our work focuses on the study of CelebA (Liu et al., 2015). Furthermore we demonstrate the efficacy of our approach on other facial datasets with more attack classes (Facescrub (Ng & Winkler, 2014)) or larger scale (VGGFace2 (Cao et al., 2018)) and on the animal dataset Stanford Dogs (Khosla et al., 2011).

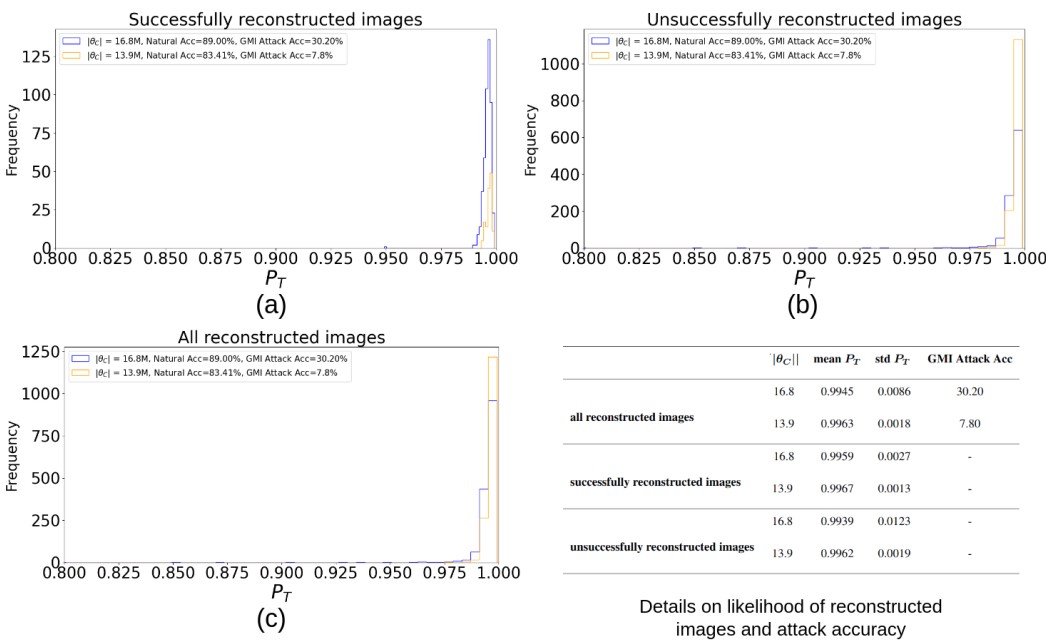

Figure 7: Visualization of the distribution of $P_T$ for two models: the no defense model (with $|\theta_C| = 16.8M$) and our proposed approach (with $|\theta_C| = 13.9M$). The visualization is conducted using GMI as the attack method, with $\mathcal{D}_{priv}$ = CelebA, $\mathcal{D}_{pub}$ = CelebA, $\mathcal{D}_{pretrain}$ = Imagenet1K, and $T$ = VGG16. We observe that both our defense model and the model without defense exhibit similar distributions of $P_T$. The values of $P_T$ for both successfully and unsuccessfully reconstructed images are very close to 1 in both cases. However, the attack accuracy shows a significant drop from 90.87% to 51.67% when our defense mechanism is applied.

**Attack Data Preparation Protocol.** Following previous works (Zhang et al., 2020; Chen et al., 2021; Wang et al., 2021a; Nguyen et al., 2023; An et al., 2022; Struppek et al., 2022) approaches, we split the dataset into private $\mathcal{D}_{priv}$ and public $\mathcal{D}_{pub}$ subsets with no class intersection. $\mathcal{D}_{priv}$ is used to train the target classifier T, while $\mathcal{D}_{pub}$ is used to extract general features only.

**Target Classifier $T$.** We select VGG16 for $T$ for a fair comparison with SOTA MI defense (Peng et al., 2022). As our approach is architecture-agnostic, we also extend the defense results on more common and recent architectures: i.e., IR152 (He et al., 2016), FaceNet64 (Cheng et al., 2017), Resnet-34, Resnet-18, Resnet-50 (He et al., 2016), ResNeSt-101 (Zhang et al., 2022), and MaxViT (Tu et al., 2022), which are not explored in previous MI defense setups (Wang et al., 2021b; Peng et al., 2022).

**Pre-trained Dataset for Target Classifier $\mathcal{D}_{pretrain}$.** We use Imagenet-1K (Deng et al., 2009) for VGG16, Resnet-18/50, ResNeSt-101, and MaxViT, and MS-CelebA-1M (Guo et al., 2016) for IR152 and FaceNe64, following previous works (Chen et al., 2021; Zhang et al., 2020). For Resnet-34, since it is trained from scratch in the original VMI setup (Wang et al., 2021a), we freeze the layers initialized from scratch. In Sec. A.3, we also study two additional pre-trained datasets, Facescrub (Ng & Winkler, 2014) and Pubfig83 (Pinto et al., 2011)

**MI Attack Method.** Our work focuses on white-box attacks, the most effective method in the literature. Following the SOTA MI defense (Peng et al., 2022), we evaluate our approach against three well-known attacks: GMI (Zhang et al., 2020), KEDMI (Chen et al., 2021), and VMI (Wang et al., 2021a). We further evaluate our approach against current SOTA MI attacks as well as other SOTA MI attacks LOMMA (Nguyen et al., 2023), PPA (Struppek et al., 2022), and MIRROR (An et al., 2022).

## D.2 EVALUATION METRICS

As mention in Sec. 4.2, we make use of Natural Accuracy, Attack Accuracy, and K-Nearest-Neighbors Distance (KNN Dist) metrics to evaluate MI robustness. These metrics are described as:

- **Attack accuracy (AttAcc).** To gauge the effectiveness of an attack, we develop an *evaluation classifier* that predicts the identities of the reconstructed images. This metric assesses the similarity between the generated samples and the target class. If the evaluation classifier attains high accuracy, the attack is considered successful. To ensure an unbiased and informative evaluation, the evaluation classifier should exhibit maximal accuracy.

- **Natural accuracy (Acc).** In addition to assessing the Attack Acc of a released model, it is also necessary to ensure that the model performs satisfactorily in terms of its classification utility. The evaluation of the model's classification utility is typically measured by its natural accuracy, which refers to the accuracy of the model in the classification problem.

- **K-Nearest Neighbors Distance (KNN Dist).** The KNN Dist metric provides information about the proximity between a reconstructed image associated with a particular label or ID, and the images that exist in the private training dataset. This metric is calculated by determining the shortest feature distance between the reconstructed image and the actual images in the private dataset that correspond to the given class or ID. To calculate the KNN Dist, an $l_2$ distance measure is used between the two images in the feature space, specifically in the penultimate layer of the evaluation model. This distance measure provides insight into the similarity between the reconstructed and the real images in the training dataset for a particular label or ID.

- $\delta_{EvalNet}$ **and** $\delta_{FaceNet}$ These metrics are measured by the squared $l_2$ distance between the activation in the penultimate layers. $\delta_{EvalNet}$ is computed via Evaluation Model while $\delta_{EvalNet}$ is computed via pre-trained FaceNet (Schroff et al., 2015). A lower value indicates that the attack results are more visually similar to the training data.

- $\ell_2$ **distance**. $\ell_2$ distance measures how similar the inverted images are to the private data by computing the distance between reconstructed features the centroid features of the private data. A lower distance means that the inverted images are more similar to the target class.

- **Frechet inception distance (FID)**. FID is commonly used to evaluate generative model to access the generated images. The FID measures the similarity between two sets of images by computing the distance between their feature vectors. Feature vectors are extracted using an Inception-v3 model that has been trained on the ImageNet dataset. In the context of MI, a lower FID score indicates that the reconstructed images are more similar to the private training images.

### D.3 MI ATTACK SETUPS

Table 9: MI Attack Setups

|  | #Iteration | #Attack ID | $w$ clipping | Learning rate | #Attack per class |
|---|---|---|---|---|---|
| **GMI** | 3000 | 300 | Yes | 0.02 | 5 |
| **KEDMI** | 3000 | 300 | Yes | 0.02 | 5 |
| **VMI** | 320 | 20 | - | 0.0001 | 100 |
| **LOMMA** | 2400 | 300 | Yes | 0.02 | 5 |
| **PPA** | 50 | 120 (Stanford Dogs), 530 (FaceScrub), 1000 (CelebA) | Yes | 0.005 | 50 |
| **MIRROR** | 500 | 100 | Yes | 0.25 | 8 |

- **GMI** (Zhang et al., 2020) uses a pre-trained GAN to understand the image structure of an additional dataset. It then identifies inversion images by analyzing the latent vector of the generator.

- **KEDMI** (Chen et al., 2021) expands on GMI (Zhang et al., 2020) by training a discriminator to differentiate between real and fake samples and predict the label as the target model. The authors also propose modeling the latent distribution to reduce inversion time and enhance the quality of reconstructed samples.

- **VMI** (Wang et al., 2021a) introduces a probabilistic interpretation of MI and presents a variational objective to approximate the latent space of the target data.

- **LOMMA** (Nguyen et al., 2023) introduces two concepts of logit loss for identity loss and model augmentation to improve attack accuracy previous MI attacks including GMI, KEDMI, and VMI.

- **PPA** (Struppek et al., 2022) proposes a framework for MI attack for high resolution images, which enable the use of a single GAN (i.e., StyleGAN) to attack a wide range of targets, requiring only minor adjustments to the attack.

- **MIRROR** (An et al., 2022) proposes a MI attack framework based on StyleGAN similar to PPA, which aims at reconstructing private images having high fidelity.

- **BREPMI** (Kahla et al., 2022) introduce a new MI attack that can reconstruct private training data using only the predicted labels of the target model. The attack works by evaluating the predicted labels over a sphere and then estimating the direction to reach the centroid of the target class.

### D.3. THE DETAILS FOR TRAINING $T$

**Training target classifier $T$.** In this work, we employ VGG16 (Simonyan & Zisserman, 2014), IR152 (He et al., 2016), and FaceNet64 (Cheng et al., 2017) for our investigation. All target classifiers are trained on CelebA dataset. For GMI (Zhang et al., 2020) and KEDMI (Chen et al., 2021), the target classifiers trained were VGG16, IR152, and FaceNet64, while Resnet-34 was used as the target classifier for VMI (Wang et al., 2021a). As mentioned in Section 4.2, we employ Imagenet-1K as the pre-trained dataset for VGG16, while MS-CelebA-1M was used as the pre-trained dataset for IR152 and FaceNet64. The details of the training procedure are shown in Tab. 10 below.

Table 10: Training settings for target classifier

| Architecture | Dataset | Input Resolution | #Epoch | Batch size | Learning rate | Optimizer | Weight Decay | Momentum |
|---|---|---|---|---|---|---|---|---|
| VGG16 | CelebA | 64x64 | 200 | 64 | 0.02 | SGD | 0.0001 | 0.9 |
| IR152 | CelebA | 64x64 | 100 | 64 | 0.01 | SGD | 0.0001 | 0.9 |
| FaceNet64 | CelebA | 64x64 | 200 | 8 | 0.008 | SGD | 0.0001 | 0.9 |
| Resnet-34 | CelebA | 64x64 | 200 | 64 | 0.1 | SGD | 0.0005 | 0.9 |
| Resnet-18 | CelebA | 224x224 | 100 | 128 | 0.001 | Adam | - | - |
| MaxViT | CelebA | 224x224 | 100 | 64 | 0.001 | Adam | - | - |
| ResNeSt-101 | Stanford Dogs | 224x224 | 100 | 128 | 0.001 | Adam | - | - |
| Resnet-50 | VGGFace2 | 224x224 | 100 | 1024 | 0.001 | Adam | - | - |

**Important Hyper-parameters.** In our work, we performed an analysis of our proposed method against existing SOTA model inversion defense methods: MID (Wang et al., 2021b) and Bilateral Dependency Optimization (BiDO)(Peng et al., 2022). MID (Wang et al., 2021b) adds a regularizer $d(x, T(x))$ to the main objective during the target classifier's training to penalize the mutual information between inputs $x$ and outputs $T(x)$. BiDO (Peng et al., 2022) attempts to minimize $d(x, z)$ to reduce the amount of information about inputs $x$ embedded in feature representations $z$, while maximizing $d(z, y)$ to provide $z$ with enough information about $y$ to restore the natural accuracy. For simplicity, we use $\lambda_{MID}$, $\lambda_x$, and $\lambda_y$ to represent $d(x, T(x))$, $d(x, z)$, and $d(z, y)$ respectively. The settings of these hyper-parameters are detailed in Table. 11 below.

## E    COMPUTE RESOURCE

All our experiments are run on a single NVIDIA RTX A5000 GPU. Given that our work is focused on model inversion defense, we provide the total training time (seconds) for the target classifier and the ratio of training time between each model inversion defense method against the No. Def. The results in Tab. 12 below show that **our proposed method can greatly reduce the amount of time required to train the target classifier.**

## F    ERROR BAR

For this section, we ran a total of 7 setups (3 times for each setup) across 4 different architectures of the target classifiers, and report their respective natural accuracy and attack accuracy values. For each experiment, we use the same MI attack setup and training settings for target classifiers as reported in

Table 11: Hyperparameters setting for training target classifiers

| Architecture | Method | $\lambda_{MID}$ | $\lambda_x$ | $\lambda_y$ | $|\theta_C|$ | Natual Acc $\Uparrow$ |
|---|---|---|---|---|---|---|
| VGG16 | No. Def | 0.01 | - | - | - | 68.39 |
| | MID | 0.01 | - | - | - | 68.39 |
| | MID | 0.003 | - | - | - | 78.70 |
| | BiDO-COCO | - | 10 | 50 | - | 74.53 |
| | BiDO-COCO | - | 5 | 50 | - | 81.55 |
| | BiDO-HSIC | - | 0.05 | 1 | - | 70.31 |
| | BiDO-HSIC | - | 0.05 | 0.5 | - | 80.35 |
| | Ours | - | - | - | 15 | 86.57 |
| | Ours | - | - | - | 13.9 | 83.41 |
| | Ours | - | - | - | 11.5 | 77.89 |
| | Ours | - | - | - | 9.1 | 69.80 |
| | Ours + BiDO-HSIC | - | 0.05 | 0.4 | 15 | 84.31 |
| | Ours + BiDO-HSIC | - | 0.03 | 0.4 | 15 | 82.15 |
| Resnet-34 | No. Def | - | - | - | 21.5 | 69.27 |
| | MID | 0 | - | - | - | 52.52 |
| | BiDO-COCO | - | 0.05 | 2.5 | - | 59.34 |
| | BIDO-HSIC | - | 0.1 | 2 | - | 61.14 |
| | Ours | - | - | - | 21.14 | 62.20 |
| IR152 | No. Def | - | - | - | 62.6 | 93.52 |
| | Ours | - | - | - | 17.8 | 86.70 |
| FaceNet64 | No. Def | - | - | - | 35.4 | 88.50 |
| | Ours | - | - | - | 34.4 | 83.61 |
| Resnet-18 | No. Def | - | - | - | 11.69 | 95.30 |
| | Ours | - | - | - | 8.91 | 91.17 |
| MaxViT | No. Def | - | - | - | 30.92 | 96.57 |
| | Ours | - | - | - | 18.31 | 93.00 |
| ResNeSt-101 | No. Def | - | - | - | 48.42 | 75.07 |
| | Ours | - | - | - | 27.94 | 79.64 |

Table 12: Computational Resource

| Architecture | Method | Total Training Time (Seconds) $\Downarrow$ | Ratio $\Downarrow$ | Natural Acc $\Uparrow$ |
|---|---|---|---|---|
| VGG16 | No. Def | 2122 | 1.00 | 89.00 |
| | BiDO-COCO | 3288 | 1.55 | 81.55 |
| | BiDO-HSIC | 3296 | 1.55 | 80.35 |
| | **Ours** | **1460** | **0.69** | **83.41** |
| | **Ours + BiDO-HSIC** | **2032** | **0.96** | **84.14** |
| IR152 | No. Def | 6019 | 1.00 | 93.52 |
| | **Ours** | **2808** | **0.47** | **86.70** |
| FaceNet64 | No. Def | 16344 | 1.00 | 88.50 |
| | **Ours** | **14448** | **0.88** | **83.61** |

Tab. 9 and Tab. 10 respectively. We show that the results obtained are reproducible and do not deviate much from the reported values in the main paper. These results can be found in Table 13 below.

## G  QUALITATIVE RESULT

**Visual Comparison.** We evaluate the efficacy of our proposed method along with BiDO for preventing privacy leakage on CelebA and also provide visualisation of the samples produced using the KEDMI (Chen et al., 2021) MI attack method. In Figure 8 below, each column represents the same identity and the first row represents the ground-truth private data while each subsequent row shows the attack samples reconstructed for each MI defense method.

**User study**. We conduct our user study via Amazon MTurk with the interface as shown above. We adapt our user study from MIRROR. In the setup, participants are presented with a real image of the target class, and then asked to pick one of two inverted images that is more closely aligned with the real image. The order is randomized, with each image pair displayed on-screen for a

Table 13: We present the results for running experiments multiples time to show the reproducibility of our proposed method. For KEDMI (Chen et al., 2021)/GMI (Zhang et al., 2020), we conduct the attacks with $\mathcal{D}_{priv}$ = CelebA, $\mathcal{D}_{pub}$ = CelebA, $\mathcal{D}_{pretrain}$ = Imagenet1K, and $T$ = VGG16/IR152/FaceNet64. For VMI (Wang et al., 2021a), we conduct the attacks with $\mathcal{D}_{priv}$ = CelebA, $\mathcal{D}_{pub}$ = CelebA, $T$ = Resnet-34, and there is no $\mathcal{D}_{pretrain}$ for this setup.

| Architecture | MI Attack | First run | | Second run | | Third run | | Average | |
|---|---|---|---|---|---|---|---|---|---|
| | | Natural Acc ⇑ | Attack Acc ⇓ | Natural Acc ⇑ | Attack Acc ⇓ | Natural Acc ⇑ | Attack Acc ⇓ | Natural Acc ⇑ | Attack Acc ⇓ |
| VGG16 | KEDMI | 83.41 | $51.67 \pm 3.93$ | 83.11 | $49.67 \pm 4.86$ | 83.54 | $53.60 \pm 4.06$ | 83.35 | $51.65 \pm 4.28$ |
| | GMI | | $7.80 \pm 3.36$ | | $8.80 \pm 2.28$ | | $8.80 \pm 3.36$ | | $8.47 \pm 3.00$ |
| Resnet-34 | VMI | 62.2 | $23.70 \pm 21.38$ | 62.88 | $19.55 \pm 12.90$ | 63.12 | $21.95 \pm 12.36$ | 62.73 | $21.73 \pm 15.91$ |
| IR152 | KEDMI | 86.7 | $64.60 \pm 4.93$ | 86.47 | $71.6 \pm 4.85$ | 86.37 | $69.33 \pm 5.03$ | 86.51 | $68.51 \pm 4.94$ |
| | GMI | | $8.93 \pm 3.73$ | | $9.47 \pm 2.57$ | | $9.60 \pm 4.16$ | | $9.33 \pm 3.49$ |
| FaceNet64 | KEDMI | 83.61 | $73.40 \pm 4.10$ | 83.01 | $76.27 \pm 4.09$ | 82.71 | $76.20 \pm 3.96$ | 83.11 | $75.29 \pm 4.05$ |
| | GMI | | $15.73 \pm 4.58$ | | $15.93 \pm 5.20$ | | $13.6 \pm 3.97$ | | $15.09 \pm 4.58$ |

Private Data

No Def.

BiDO-HSIC

Ours

Ours + BiDO-HSIC

Figure 8: Qualitative results to showcase the effectiveness of our proposed method, using KEDMI (Chen et al., 2021) with $\mathcal{D}_{priv}$ = CelebA, $\mathcal{D}_{pub}$ = CelebA, $\mathcal{D}_{pretrain}$ = Imagenet1K, and $T$ = VGG16. The visual comparison reveals that our proposed method achieves competitive reconstruction of private data, while the hybrid approach combining our method with BiDO-HSIC demonstrates a significant degradation in MI attack and reconstruction quality.

maximum duration of 60 seconds. The assessment encompassed all 300 targeted classes. Each pair of inverted images is assigned to 10 unique individuals, thus our user study involves a total of 3000 pairs of inverted images. We use KEDMI as the MI attack with $\mathcal{D}_{priv} = CelebA$, $\mathcal{D}_{pub} = CelebA$, $T = FaceNet$. *Consistent with the AttackAcc, the user study shows that our method provides better defense against the reconstruction of private data characteristics compared to BIDO.*

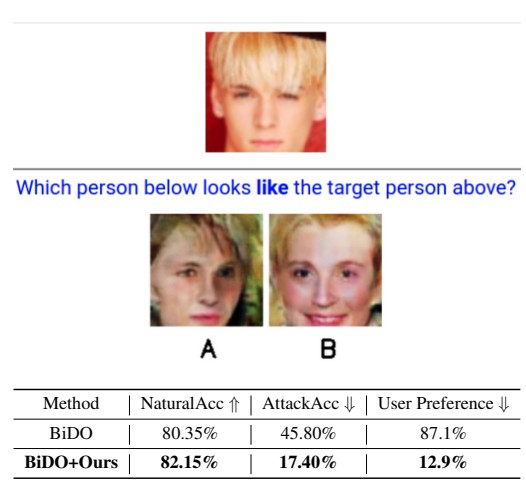

| Method | NaturalAcc ⇑ | AttackAcc ⇓ | User Preference ⇓ |
|---|---|---|---|
| BiDO | 80.35% | 45.80% | 87.1% |
| **BiDO+Ours** | **82.15%** | **17.40%** | **12.9%** |

Figure 9: User study results

