# OpenReview forum: "Model Inversion Robustness: Can Transfer Learning Help?"
_ICLR.cc/2024/Conference — ICLR 2024 Conference Withdrawn Submission_

### Official Review · Reviewer_zxaM · 2023-10-23

**Soundness:** 3 good
**Presentation:** 3 good
**Contribution:** 1 poor
**Rating:** 3
**Confidence:** 5

**Summary:**

In this work authors propose a novel defence against model inversion attacks based on limiting the information flow within the model by restricting the layers which are trained on the sensitive data by using transfer learning.

**Strengths:**

The work is structured very well, has numerous figures and has a source code attached. The evaluation is quite extensive showing the advantages of the proposed method.

**Weaknesses:**

I do, however, have a number of concerns.

Firstly I do not see anything scientifically new proposed here. It was shown before that A) certain layers can ‘memorise’ more useful information to aid a MI attacker [1] and B) that reducing the number of layers in the model can reduce the amount of useful information leant and hence limit attacker’s capabilities [2] (which does not universally hold either, but more on that later). It is also important to note that this work claims all of its results and conclusions in a generic MI setting, which is not fully true.

Take gradient-based MI, for instance: Having smaller models (or parts of models) that are exposed to sensitive data, is arguably an easier setting for such an attacker to exploit. Similarly, other works on generative MI have previously shown the opposite results to what you share here [3] - when such mitigation (in that case split learning) is present, it is actually beneficial for the attacker to use a smaller section of the model (due to the ease of reconstruction). I do appreciate that these settings are different to the one described in this work, but the method proposed here does not apply to any generic MI setting, contrary to what authors are trying to suggest.

Moreover, the entire concept of information useful for inversion attacks (as well as how to reduce it) was previously studied in [1], showing that based on the nature of the attack, different layers can contain various amounts of information useful for each specific adversarial method. Thus, I cannot see any novelty in the FI analysis either, as the conclusions are hardly new.

Besides, this method is based on the fact that some public dataset exists, allowing to ‘outsource’ a lot of the learning to data which is not deemed to be sensitive. This is A) a relatively strong assumption for a generic MI defence and B) has been previously proposed as a potential solution against inference attacks already in [4] and the baselines discussed in that work. Point A) has already been somewhat discussed in work on differentially private learning, showing that yes, it is a valid idea, but it a strong assumption to make (i.e. more public data means there is less private data required and, hence, the privacy budgets can be smaller).

**Questions:**

How can this method be scaled to other types of inversion attacks? Is there any correlation between information that can be used by MI attacker and, for instance, membership or attribute inference attacker?

Overall, this work really lacks scientific novelty and is limited to only a subset of MI attacks, making it severely less applicable to realistic ML settings. In my view, the answer to the question ‘can transfer learning help against MI?’ has previously been answered in other works.


[1] - Mo, Fan, et al. "Quantifying information leakage from gradients." CoRR, abs/2105.13929 (2021).
[2] - Wang, Yijue, et al. "Against membership inference attack: Pruning is all you need." arXiv preprint arXiv:2008.13578 (2020).
[3] - Usynin, Dmitrii, et al. "Zen and the art of model adaptation: Low-utility-cost attack mitigations in collaborative machine learning." Proc. Priv. Enhancing Technol. 2022.1 (2022): 274-290.
[4] - Chourasia, Rishav, et al. "Knowledge Cross-Distillation for Membership Privacy." arXiv preprint arXiv:2111.01363 (2021).

---

### Official Review · Reviewer_TtAY · 2023-10-27

**Soundness:** 3 good
**Presentation:** 3 good
**Contribution:** 3 good
**Rating:** 6
**Confidence:** 5

**Summary:**

The paper proposes a new defense strategy against model inversion (MI) attacks by limiting the number of parameters fine-tuned on private data using transfer learning. In specifc, the method involves pre-training on public data, then fine-tuning only the last few layers on private data, freezing the first layers to prevent private data encoding. The paper analyzes layer importance for MI task using Fisher information, suggesting first few layers are important for MI, while last several layers are import for classification task. Experiments show the method achieves state-of-the-art MI defense across various setups and architectures, while being simple to implement.

**Strengths:**

originality: This work proposes an innovative approach to defending against MI attacks, which opts for transfer learning over traditional dependency based regularization methods. The idea is not only novel but intuitively appealing, offering a fresh perspective on the problem. What further strengthens the argument is the incorporation of Fisher information, which provides a robust justification for the proposed method. This logical integration of Fisher information bolsters the paper's overall coherence and the credibility of the proposed approach
quality: This work conducts extensive array of necessary experiments, consistently attains state-of-the-art performance across various model inversion setups and architectural configurations. The comprehensive and rigorous experimental evaluation conducted in this work underscores its significant contributions to the field.
clarity: The paper is well-written and easy to follow logically.
significance: A new perspective on MI defense.

**Weaknesses:**

Potential misleading claim:
"In other words, MID and BiDO reduce MI attack accuracy by suppressing likelihood P(y|x)." it would be beneficial to clarify whether there is experimental evidence supporting the claim that MID and BiDO reduce MI attack accuracy by suppressing the likelihood P(y|x). If such experiments have been conducted, providing references or details about them would strengthen the paper's credibility. If not, it may be advisable to rephrase this statement as a hypothesis or a potential outcome to avoid overclaiming the results.

Unaligned evaluation metrics:
The evaluation metrics for 'AttAcc' across different MI attacks are not consistent. For instance, in experiments involving GMI, KEDMI, and VMI, the reported 'AttAcc' values are accompanied by standard deviations, which typically indicate variations in 'AttAcc' across different classes, rather than the results of multiple experiment runs. Conversely, in the case of PPA and MIRROR, only 'AttAcc' is provided, which lacks the same level of rigor. It would be helpful to standardize the reporting of 'AttAcc' to maintain consistency in the evaluation process. Additionally, providing clarity on the interpretation of standard deviations in GMI, KEDMI, and VMI experiments, such as whether they represent variations across classes or multiple experiment runs, would enhance the paper's transparency.

**Questions:**

1. For experiments related to VMI, it raises a valid concern that there is no D_pretrain. This absence of a pretraining backbone questions the relevance of these experiments in demonstrating the effectiveness of your proposed method. It would be beneficial to provide a clear rationale for including VMI experiments in the evaluation.

---

### Official Review · Reviewer_8Fh6 · 2023-10-28

**Soundness:** 3 good
**Presentation:** 3 good
**Contribution:** 2 fair
**Rating:** 5
**Confidence:** 4

**Summary:**

The paper introduces a defense mechanism against Model Inversion (MI) attacks, which threaten privacy by reconstructing private training. Instead of traditional regularization techniques, the paper employs transfer learning to restrict the encoding of sensitive information to specific layers, hindering MI attacks. An analysis using Fisher Information reveals the importance of the initial model layers for MI attacks, supporting the proposed defense approach. Empirical validation demonstrates that reducing the number of fine-tuned parameters in private datasets significantly reduces MI attack accuracy.

**Strengths:**

- Interesting Defense Approach: The paper introduces a new defense strategy using transfer learning.
- Fisher Information Analysis: The paper conducts a unique analysis of layer importance using Fisher Information, providing valuable insights.
- Practicality: The proposed defense is straightforward to implement and can be applied to a wide range of model architectures.

**Weaknesses:**

- Limited Attack Coverage: The defense's focus on attacks leveraging GANs may not address all model inversion attack types.
- Unclear Threat Model: The paper lacks clarity in specifying the adversary's knowledge of critical elements, which affects the practicality of the defense.
- Performance Not Satisfying: The defense's performance is questionable, as it doesn't significantly outperform existing methods.

**Questions:**

Three significant concerns merit attention in the evaluation of the paper.

Firstly, while the authors assert the applicability of their defense against general model inversion attacks, it becomes evident from the background section that their primary focus pertains to attacks leveraging extra information from GANs. This orientation does not account for the broader scope of general model inversion, as several attacks rely solely on the model's parameters. It is advisable for the authors to undertake discussions and experimentation that substantiate the efficacy of their defense in mitigating vulnerabilities across a wider spectrum of attack types.

The foremost concern that arises pertains to the lack of a clear threat model. It remains essential to delineate whether the adversary possesses knowledge of critical elements such as the pretrained model, the pretrained dataset, and the fixed layers. Without such clarification, it becomes difficult to assess the practicality of the proposed defense, particularly, if the adversary has knowledge of the pretrained model and final model, they can perform the model inversion attack as indicated in Salem et al.’s work [1].

Attack performance is not satisfying. In particular, the results presented in Table 5 demonstrate that, when compared with BiDO, the proposed defense does not exhibit a significant advantage. A more meaningful approach to comparison might involve holding the attack accuracy constant while comparing the original accuracy or vice versa. Notably, the proposed defense leads to a 3% increase in accuracy but also results in a 5% increase in attack accuracy, raising questions about its superiority over previous defense methods. This consideration becomes particularly salient given the heightened complexity of the threat model associated with the proposed defense in comparison to earlier approaches.

[1] Updates-Leak: Data Set Inference and Reconstruction Attacks in Online Learning

---

### Official Review · Reviewer_LRKJ · 2023-10-31

**Soundness:** 3 good
**Presentation:** 3 good
**Contribution:** 2 fair
**Rating:** 5
**Confidence:** 4

**Summary:**

The paper proposes to use transfer learning as a possible defense against model inversion attacks. The proposed approach makes use of pre-trained models and only fine-tunes part of the parameters. Using Fisher Information, the importance of the first few layers for successful model inversion attacks is shown, while for the classification task itself the first few layers are not very important. The experimental results show that fine-tuning fewer layers seems to reduce the attack success of model inversion attacks, while at the same time having only moderate impact on the accuracy of the model. The proposed approach is compared to other existing model inversion defense methods (BIDO and MID) and it is shown that the proposed approach achieves lower attack success while at the same time having less reduction in accuracy. The main finding of the paper is that fine-tuning fewer parameters on a private dataset, the model memorizes/leaks less private information about the training data.

**Strengths:**

- The paper is well-structured and easy to follow
- The topic of the paper is an interesting and important area of research
- Even though the proposed method is merely an application of an existing technique, using transfer learning for defending against model inversion attacks seems to be novel

**Weaknesses:**

- the different number of parameters are not visible in Fig. 1-IV
- Table 3: does only show the proposed defense against no defense and the proposed approach is not compared to existing defense methods
- Tables in the appendix are referenced in Section 4.5. This is not a good practice to reference tables in the appendix as if they were in the main part of the paper
- from the paper, the whole attack scenario is not quite clear. It is not specified how many classes are attacked and how these classes are chosen.
- I suspect to measure the Attack Accuracy, a validation model was trained. However, it is not stated how this validation model was trained or what architecture this validation model has.
- even though the proposed approach is compared to other defenses using the GMI and KEDMI attack, a comparison to other defenses on other attacks like PPA or MIRROR is missing.

Misc:
- missing "s" in introduction "MI attack is a type of privacy threat that aim[s]"
- Page 6: missing plural "The set of MI reconstructed images [...] for different identit[ies] is used as X"
- Page 7: missing plural "[...] when we combine [the] two approach[es]"
- Page 8: missing "s" "Therefore, our approach offer[s] [...]"
- Page 9: missing word "[...], regardless [of] multiple factors [...]"

**Questions:**

- Q1: How often are the experiments repeated?
- Q2: What are the dots in Fig. 1-IV?
- Q3: Could you provide a comparison of the approach to MID and BIDO on PPA and MIRROR?

**Details Of Ethics Concerns:**

No ethics review needed since the paper only proposes a defense against model inversion attacks.